# Teratoid Hepatoblastoma—Our Experience

**DOI:** 10.3390/cancers14246135

**Published:** 2022-12-13

**Authors:** Lara Berklite, Sarangarajan Ranganathan

**Affiliations:** Department of Pathology, Cincinnati Children’s Hospital Medical Center, 3333 Burnet Avenue, Cincinnati, OH 45229, USA

**Keywords:** pediatric, hepatoblastoma, teratoid, liver

## Abstract

**Simple Summary:**

Hepatoblastomas (HB) are the most common pediatric liver tumor with a number of histologic patterns described. Teratoid HB is a rare subtype with only a few case reports documented in the current literature. The aim of this study is to analyze a large series of teratoid HB to better characterize the clinicopathologic features in order to aid in recognition and diagnosis, improve understanding of the biology of these tumors, and provide insights into clinical behavior that may inform future clinical trials.

**Abstract:**

Hepatoblastomas (HB) are the most common pediatric liver tumor with several subgroups described, of which teratoid HB is the rarest. The aim of this study is to characterize the histologic and phenotypic spectrum of teratoid HB in order to better understand the biology and behavior of these tumors. A retrospective analysis of all teratoid HB diagnosed at a major pediatric hospital as well as the consultation files of one of the authors (SR) was performed with the available clinical data and surgical pathology material reviewed. A detailed immunohistochemical workup was also performed. A total of 28 cases were included from patients ranging from 5 to 84 months of age and a M:F ratio of 1.07:1. Four patients had syndromic associations. In 14/28 cases, the tumors contained primitive glandular elements with histologic and immunophenotypic overlap with the yolk sac tumor which in two cases became predominant in metastatic sites. One case had extensive primitive neural epithelium mimicking a primitive neuroectodermal tumor (PNET). Other unique elements included melanin, mature neuroglial tissue, rhabdomyoblastic differentiation, and neuroendocrine carcinoma-like areas (*n* = 2). In conclusion, this study provides the largest series of teratoid HB to date with clinical and outcome data, highlights previously undescribed or under-recognized histologic patterns in these tumors, and describes the immunohistochemical profile of these tumors to aid in diagnosis.

## 1. Introduction

Hepatoblastomas (HB) are the most common pediatric liver tumors with an annual incidence of 10.5 cases per million children less than 1 year of age (SEER database). The new consensus classification has described the histology of various subgroups of HB [1]. Teratoid HB is the least common subtype with no definite incidence known for this variant as only a small number of teratoid HB have been reported in the literature, mostly as isolated case reports [2]. By definition, the presence of primitive neuroepithelial elements or melanin is required to qualify for the diagnosis, but detailed clinical, pathologic, and immunohistochemical analyses of teratoid HB are lacking [1]. Recognition of the full spectrum of features that may be seen in these tumors is imperative to avoid overlooking a minor teratoid component which may become predominant in later specimens. In small biopsies with a significant teratoid component, failure to recognize the features as indicative of a hepatoblastoma could result in misdiagnosis of a germ cell tumor. In addition, the prognostic and biologic significance of the presence of teratoid elements is unknown and better characterization of teratoid HB is necessary to improve such knowledge. This study was conducted with the aim of characterizing the histologic and phenotypic spectrum of teratoid HB. Unusual variants and presentations not previously described are also illustrated.

## 2. Materials and Methods

A retrospective analysis of all hepatoblastomas seen at a large tertiary pediatric hospital, either as surgical or personal consults, was reviewed after IRB approval. Cases were excluded if surgical pathology material was not available or if the histologic features were insufficient to meet current criteria for the designation of teratoid hepatoblastoma as described in the most recent consensus classification. Detailed immunohistochemical workup of cases was reviewed with special emphasis on staining for beta-catenin, glypican-3, glutamine synthetase, MOC31, Prox1, and SALL4, with adequate controls. Details for these immunohistochemical stains are included in Table 1. Additional stains were performed where needed. Relevant clinical data were collected from the electronic medical record, including presenting symptoms, AFP levels, medical intervention and treatment history, and clinical follow-up when available.

## 3. Results

After retrospective review of previously diagnosed hepatoblastomas over a 42-year period, a total of 28 cases were confirmed as containing teratoid elements and included in the current series. Five cases were excluded from the series as they failed to meet the diagnostic criteria, with two cases only containing pigment equivocal for melanin, one case demonstrating only patchy expression of neuroendocrine markers by immunohistochemistry, and two cases failing to demonstrate any reproducible histologic teratoid features.

### 3.1. Clinical

The age range was 5 months to 84 months and the male to female ratio was 1.07:1 (Table 2).

Alpha-fetoprotein (AFP) levels were documented as elevated at the time of presentation in 20 patients with available results. AFP levels ranged from 3183 to 1,260,000 ng/mL. Four patients had syndromic associations including Beckwith–Wiedemann syndrome (*n* = 3), trisomy 13 (*n* = 1), and Wolf–Hirschhorn syndrome (*n* = 1). Two patients had a documented history of prematurity. One 2-year-old male patient had an unusual presentation with precocious puberty and elevated AFP, testosterone, 17-OH progesterone, and beta-HCG (THB25). Six patients underwent liver transplants and nineteen patients had surgical resections. In the remaining patients, clinical data were limited as only biopsy specimens were received in consultation. Eight patients had lung, abdominal lymph node, peritoneal, or omental metastases. Since the early 2000s, most patients have been treated on standard Children’s Oncology Group protocols for these tumors determined initially by their stage at presentation and ability to resect upfront and more recently on the latest protocol based on PRETEXT staging. Details of all treatments for individual patients are not available for the purposes of this study. Three patients have died of disease. One patient had a disease-free survival of 14 years but died due to renal failure at 18 years of age (THB3). Two patients had documented recurrence or progression during a follow-up period of 2 years (THB16, THB23). The other patients were well at varying follow-up periods (1 month–25 years).

### 3.2. Pathology

#### 3.2.1. Histology

The majority of tumors were mixed epithelial and mesenchymal hepatoblastomas (75%, 21/28). The epithelial components were most commonly composed of well-differentiated fetal (*n* = 14), crowded fetal (*n* = 21), embryonal (*n* = 24), and blastemal (*n* = 13) elements. A few cases also had cholangioblastic (*n* = 8), pleomorphic fetal (*n* = 6), and macrotrabecular patterns (*n* = 3). Mesenchymal components were typically composed of spindled cells or osteoid/bone with a few tumors containing areas of skeletal muscle/rhabdomyoblastic (*n* = 3, Figure 1A,B) or smooth muscle differentiation (*n* = 1). The spectrum of teratoid elements are illustrated in Figure 1. Of the teratoid elements identified, one of the most common was melanin pigment (*n* = 10), often in association with squamous differentiation with or without keratin formation (Figure 1C). These foci of melanin and squamous differentiation were sometimes located centrally within epithelial tumor islands with fetal morphology. In other cases, the melanin positive cells were intimately associated with osteoid (THB14, THB16, THB21). In 4/28 cases, melanin was the only teratoid component identified within the tumor and in some cases was very focal, only represented in the resection specimen and not present on initial biopsy (THB16, THB21). In a few of these cases significant post-therapy changes were present with abundant hemosiderin deposition in addition to melanin, and an iron stain was helpful to confirm the diagnosis (THB 21, Figure 1D). Other teratoid elements identified included neural elements with a range of maturation including mature neuroglial tissue, ganglion cells, and primitive neuroepithelium (*n* = 15, Figure 1E–H). One case was unique in that it had extensive primitive neuroepithelium with PNET-like areas which made the diagnosis of hepatoblastoma challenging (THB17, Figure 2).

A significant number of teratoid tumors contained glandular components (*n* = 16). In a few cases (THB18, THB24) the glandular components demonstrated differentiation, recapitulating gastrointestinal or mature mucinous epithelium, but in many cases the glands appeared primitive, defying definitive classification. These primitive glandular components often had subnuclear and/or supranuclear vacuoles and a subset had significant morphologic overlap with yolk sac tumors. A few rare cases contained neuroendocrine-like elements (THB12, THB25, Figure 3).

In those patients with metastasis, four had teratoid elements present at distant metastatic sites (THB12, THB15, THB20, THB23) consisting of ganglion cells, melanin, primitive glands or yolk-sac-tumor-like elements. In two cases, distant metastases were composed entirely of yolk-sac-tumor-like elements (THB12, THB23). One case initially had omental and extrahepatic tumors with mixed fetal, embryonal, and teratoid elements (Figure 4) and subsequently developed peritoneal metastases exclusively composed of tubulopapillary yolk-sac-tumor-like areas (Figure 5A,B, THB23).

The second case has been previously reported [3] and showed a mixed mesenchymal and epithelial hepatoblastoma at the time of resection with fetal and embryonal epithelial elements and osteoid as well as areas with insular to trabecular growth resembling a neuroendocrine tumor located between the osteoid. Subsequently, following salvage therapy the patient developed liver recurrence, retroperitoneal lymphadenopathy, and a perirenal mass which on biopsy were all composed of yolk-sac-tumor-like elements.

#### 3.2.2. Immunohistochemistry

Primitive glandular and yolk-sac-tumor (YST)-like elements demonstrated nuclear positivity for nuclear SALL4 (10/11) and beta-catenin (9/11), although in two cases the nuclear beta-catenin was only focal (THB12, THB23) (Figure 5C–H). The primitive glandular/YST-like elements also frequently showed membranous positivity for MOC31 (7/10), although this was sometimes found in only a subset of the glandular elements (THB25, THB28). In 8/11 cases, the primitive glandular/YST-like elements demonstrated focal cytoplasmic glypican-3. These elements were largely negative for glutamine synthetase with a few cases (4/10) showing focal weak cytoplasmic expression (THB8, THB11, THB15, THB23). PROX-1 was also mostly negative (5/8) in these elements. In contrast, glandular elements with evidence of differentiation along gastrointestinal lineage did not show staining for SALL4 or demonstrate nuclear beta-catenin (Figure 2E–H, THB18, THB24). Neural elements also showed predominantly cytoplasmic-to-membranous expression of beta-catenin with only rare nuclei staining (1/5), including in cases with more primitive elements. The staining pattern for SALL4 in neural elements largely reflected the degree of differentiation with mature elements being entirely negative and only a few cases (*n* = 3, Figure 2G) with very primitive elements showing some nuclear positivity. Neural elements were consistently negative for glutamine synthetase and glypican-3. MOC31 was variably positive in neural elements with two of four cases with available material for immunohistochemistry showing some expression and two of four cases being entirely negative. Synaptophysin highlighted neuroendocrine (Figure 3, THB12, THB25, THB27) and mature neuroglial areas with rare positive cells within glandular elements (THB14, THB15). Many cases additionally had small clusters of primitive embryonal cells with synaptophysin positivity (THB16, THB20, THB21, THB22, THB23).

## 4. Discussion

Teratoid hepatoblastomas (HB) are an unusual but enigmatic group of neoplasms that show differing patterns with only rare reported cases in the literature (Table 3).

Teratoid HB was first described by Manivel et al. in reference to a tumor in an 8-month-old girl which demonstrated the unique presence of neuroepithelial-like embryonic tubules, melanocytes, and ganglion cells [6]. The authors of that paper postulated that the presence of these unusual elements was the result of neuroectodermal differentiation of multipotential neoplastic cells [6]. The authors noted that while other heterologous elements had been previously reported in hepatoblastoma, including squamous differentiation, primitive tubules or ducts recapitulating developing intestinal or respiratory epithelium, smooth muscle, cartilage, bone, and skeletal muscle, these were, in contrast, all endodermal-derived elements [6]. This is the largest reported series of teratoid HB in the current literature and the first study that highlights unique phenotypes and detailed IHC profile of these unusual tumors. Our series highlights that teratoid HB may not necessarily show neuroepithelial or melanin components in all cases and can have unusual clinical features.

While the more widely recognized teratoid elements of melanin (10/28) and variably differentiated neural elements (15/28) were present in many of our cases, a significant number (14/28) contained primitive glands with overlapping features with yolk sac tumors, indicating this is likely an under-recognized pattern that should be carefully evaluated for when reviewing an HB. In two of our cases, this pattern became predominant in subsequent metastases underscoring how recognition is imperative to avoid a misdiagnosis. It seems likely based on our study that previous reports in the historical literature of cases of “mixed HB and teratoma” [8,11,16] may actually represent additional examples of this unusual pattern of HB and add support to the idea that this pattern can cause diagnostic confusion. Histologically, these YST-like components can have a variety of appearances ranging from microcystic to glands with subnuclear vacuolization. In our study, we have demonstrated that IHC can be a useful adjunct in identifying YST-like elements and differentiating them from more differentiated glandular elements or neuroepithelium. These areas often demonstrate positivity for nuclear beta-catenin, glypican3, SALL4, and MOC31 in contrast to mature glands and neuroepithelium where positivity for these markers is absent in the case of mature glands or less consistently expressed, as in many neural elements in the current study. The underlying mechanism driving the evolution to a predominantly yolk-sac-tumor-like morphology and phenotype is unknown but the presence of nuclear beta-catenin supports a shared molecular pathogenesis with conventional HB. Considering the embryologic origin of the liver from the celomic cavity (yolk sac), it is not difficult to postulate that the presence of these primitive yolk sac elements may represent dedifferentiation of the stem cells that give rise to HB. This also reflects the totipotent capability of the HB stem cells. Further study is needed to identify whether these elements harbor additional molecular alterations that allow them to become dominant in the post-chemotherapy setting. In at least two cases, the tumors were particularly aggressive, but whether presence of YST elements in a primary HB tumor portends a poorer outcome requires additional investigation.

Other unusual and unique histologic patterns identified in our series included one case with a predominance of very primitive PNET-like neural elements and two cases with neuroendocrine elements. One of the cases with neuroendocrine elements had a striking clinical presentation with precocious puberty and elevated levels of testosterone, 17-OH-progesterone, and beta-hCG in addition to elevated AFP. Prior to this case, there is only one previously reported case of HB presenting with endocrine manifestations, occurring in a 6-year-old female who had Cushing syndrome and whose liver tumor was found to be secreting ACTH, CRH, and PTHrP [17]. Unfortunately, this case report did not include a detailed histologic description so it is unknown if this case contained teratoid elements. A few studies have investigated the expression of neuroendocrine markers including synaptophysin, chromogranin, serotonin, and CD56 in HB by immunohistochemistry [18,24] describing clusters of positive cells in epithelial components with fetal and embryonal morphology as well as focal expression within glandular components. In our series, synaptophysin highlighted mature neural elements, showed focal and weak positivity in the glandular components of a few cases, and highlighted neuroendocrine elements. Notably, many of the cases in the current series without histologic evidence of neuroendocrine differentiation showed scattered small clusters of synaptophysin-positive cells which had morphology consistent with a primitive embryonal component. This finding illustrates that caution must be used when diagnosing a neuroendocrine component within an HB and the designation should be reserved for those cases in which both morphologic and immunophenotypic neuroendocrine features are present and not on focal expression of neuroendocrine markers alone. Interestingly, previous studies have demonstrated that subsets of hepatoblasts and the more primitive liver stem cells in the developing liver express neuroendocrine markers including NCAM and chromogranin [25]. We have also noted expression of synaptophysin within embryonal areas of HB. Synaptophysin and chromogranin are also expressed within the developing ductal plate, the site of the development of the intrahepatic bile ducts [25].

Whether teratoid elements originate from a population of multipotent or less committed stem cells remains an open question. In general, hepatoblastoma is hypothesized to arise either from primary hepatoblasts or a multipotent hepatic progenitor cell [26]. Embryologically, there is evidence of an early bipotential population of progenitor cells in the foregut endoderm which can go on to participate in either liver or pancreas development [27]. The liver primordium or liver bud which develops as hepatic cords grow into the mesenchyme of the transverse septum [27] contains multipotential cells that represent the most primitive stem cells of the liver. These stem cells undergo further differentiation to produce hepatoblasts which demonstrate expression of CK8, CK18, CK7, CK19, AFP, and glypican-3 [25] as well as MOC-31 and PROX-1 and exhibit bipotential to differentiate into both hepatocytes and cholangiocytes. The origin of hepatic stem cells is still debated, but evidence suggests that they may be derived from the bone marrow as studies in mouse models have demonstrated that purified hematopoietic stem cells have the capability to differentiate into mature hepatocytes [28]. Within the adult liver there are a subset of cells termed hepatic progenitor cells with overlapping molecular characteristics with primitive hepatoblasts which may be activated in severe injury to the liver [25]. More recently, a population of cells was identified within peribiliary glands in adults and infants with overlapping features to the stem cells identified in embryonic development and which were demonstrated to be capable of differentiating into intestinal and pancreatic epithelial cells, fat, bone, cartilage, and endothelial cells [25]. The presence of stem-like cells in hepatoblastoma, which might explain the presence of teratoid elements, has undergone much investigation. Ruck et al. identified a population of “small epithelial cells” with features by immunohistochemistry and ultrastructure which were intermediate between hepatocytes and cholangiocytes [10]. Zimmerman et al. also described a subset of so-called “immature-looking cells” which showed stronger beta-catenin expression, decreased Ki67 proliferative index and lower expression of hepatocyte antigens [27]. These descriptions likely correspond to those “small cell undifferentiated” elements now designated as “blastema” within hepatoblastoma. Some have proposed that these small epithelial cells resemble hepatic stem cells. Interestingly, larger areas of blastema are reportedly present in teratoid HB [2]. In our series, blastemal areas were present in 13/28 cases and in 9/16 cases with glandular components.

It has been suggested that teratoid elements may be less chemo-responsive given observations that in post-chemotherapy specimens, these elements may predominate as the more chemosensitive epithelial hepatoblastoma components regress [21]. Corroborating this idea is the fact that in our series the teratoid components were seen at metastatic sites in four cases, including two cases, one previously reported, where the metastases were composed entirely of yolk-sac-tumor-like elements [3]. Other reports have suggested that chemotherapy may induce maturation of teratoid elements [5,6,13,15]. This assertion is difficult to assess based on our series as the full spectrum of teratoid elements was often not represented on the small biopsies received prior to chemotherapy. As previously noted, more study is needed to further clarify the issue of prognostication in these rare tumors.

## 5. Conclusions

In conclusion, this study expands the current knowledge on the wide variety of patterns seen in teratoid HB, providing tools to the practicing pathologist to aid in recognition and to facilitate further investigation into the biology of these rare tumors. These are enigmatic tumors that almost always occur in the setting of classic mixed epithelial and mesenchymal hepatoblastoma but may rarely be noted in only the context of an epithelial hepatoblastoma. They highlight the multipotent capability of the HB stem cell, which is different than the stem cells seen in adult tumors which do not show this multidirectional dedifferentiation or lineages. This finding is similar to that seen in Wilms tumors and may suggest similar biology for these pediatric tumors despite their distinct molecular pathogenesis. It is interesting to note the association with BWS in these patients is similar to Wilms tumors.

The fact that nuclear beta-catenin is a feature in many of the primitive areas suggests the role of the Wnt/beta-catenin pathway in pathogenesis, more mature elements show less evidence of nuclear beta-catenin staining and may represent a more metaplastic differentiation in these tumors. Further molecular studies are in progress to better understand the pathogenesis of these tumors, which constitute less than 2% of all hepatoblastmas.

## Figures and Tables

**Figure 1 cancers-14-06135-f001:**
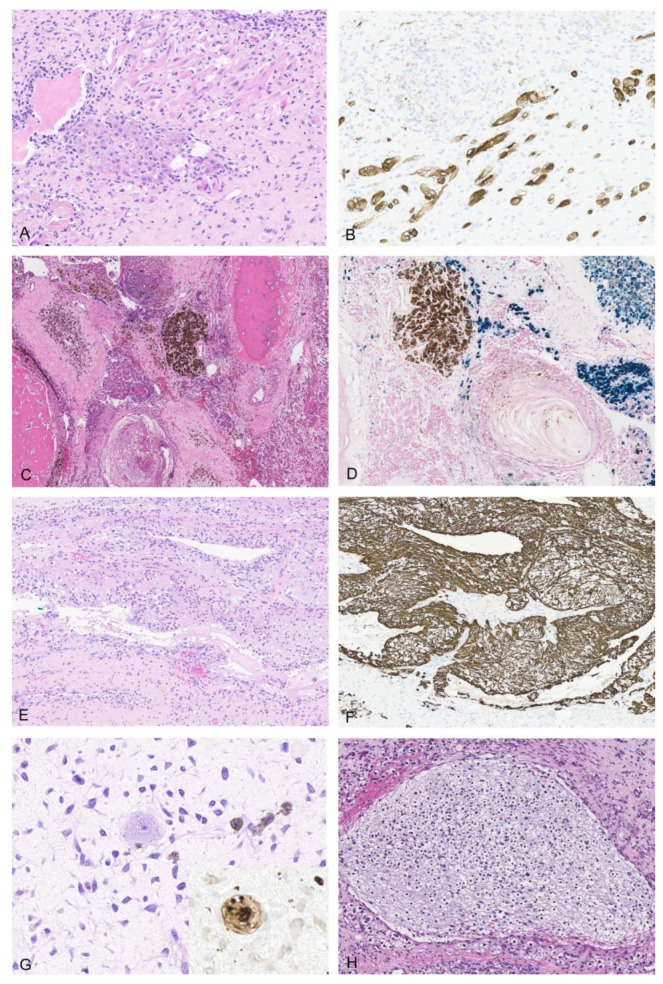
Rhabdomyoblastic differentiation, neuroglia, and melanin in teratoid HB. (**A**) Rhabdomyoblastic differentiation in THB14 (H&E 10×). (**B**) Desmin IHC in THB14 10×. (**C**) Melanin in association with squamous differentiation with post-therapy effects in THB21 (H&E 4×). (**D**) Iron stain in THB21 10×. (**E**) Mature neuroglial elements in THB14 (H&E 10×). (**F**) GFAP IHC in THB14 10×. (**G**) Ganglion cells in THB14 (H&E 20×) and PGP9.5 IHC in THB14 20× (Inset). (**H**) Mature neuroglial elements in THB18 (H&E 10×).

**Figure 2 cancers-14-06135-f002:**
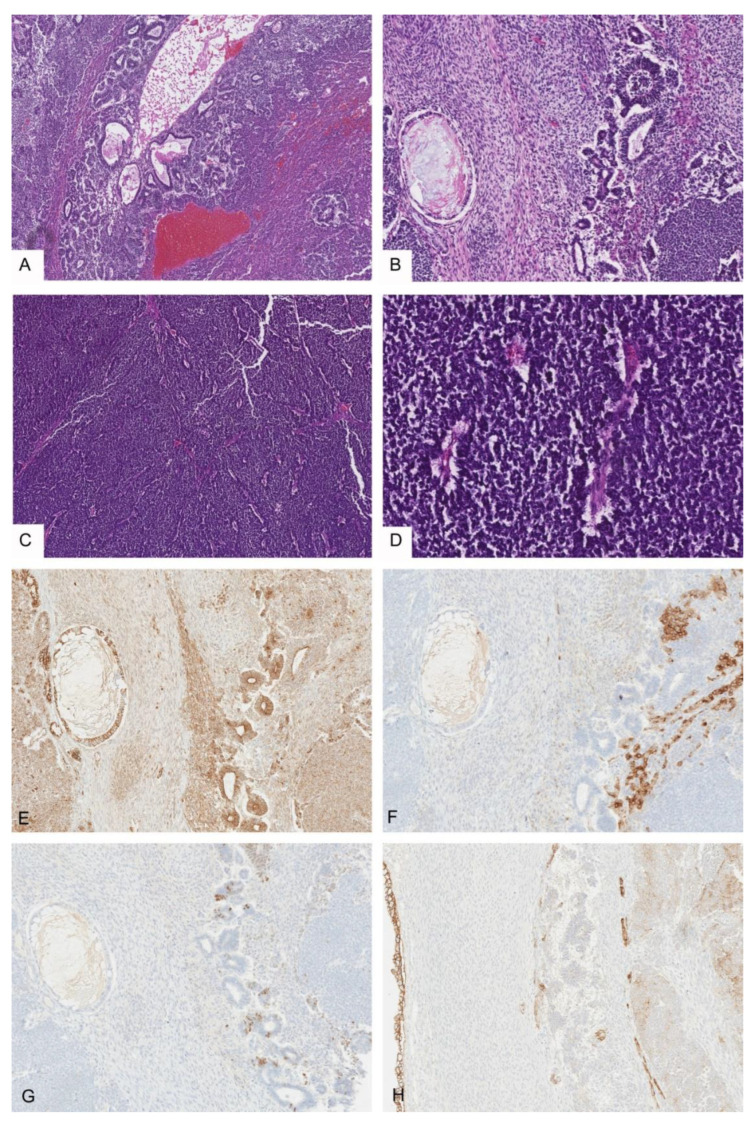
Teratoid HB with extensive primitive neuroepithelium and PNET-like areas. (**A**,**B**) THB17 with primitive neuroepithelium, variably differentiated glandular elements, 4×, 10×; (**C**,**D**) extensive PNET-like areas in THB17, 4×, 20×; (**E**) beta-catenin IHC 10×; (**F**) glypican-3 IHC 10×; (**G**) SALL4 IHC 10×; (**H**) MOC31 IHC 10×.

**Figure 3 cancers-14-06135-f003:**
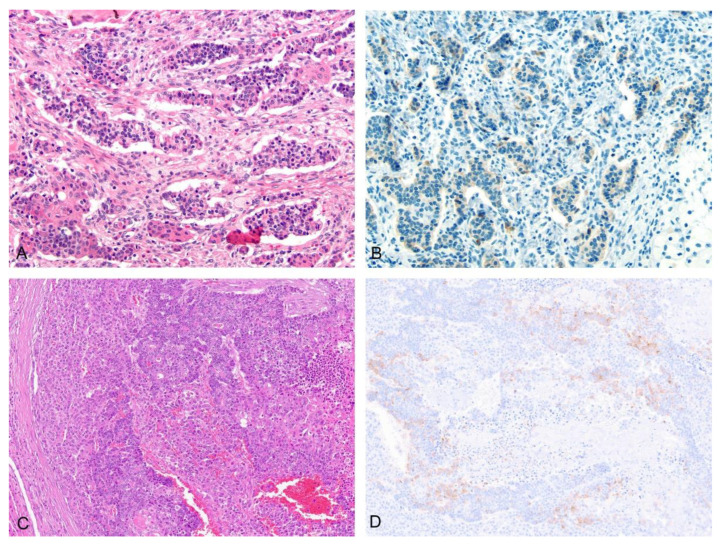
Neuroendocrine-like elements in THB12 and THB25. (**A**) H&E THB12 20×; (**B**) Synaptophysin IHC THB12 20×; (**C**) H&E THB25 10×; (**D**) Synaptophysin IHC in THB25 10×.

**Figure 4 cancers-14-06135-f004:**
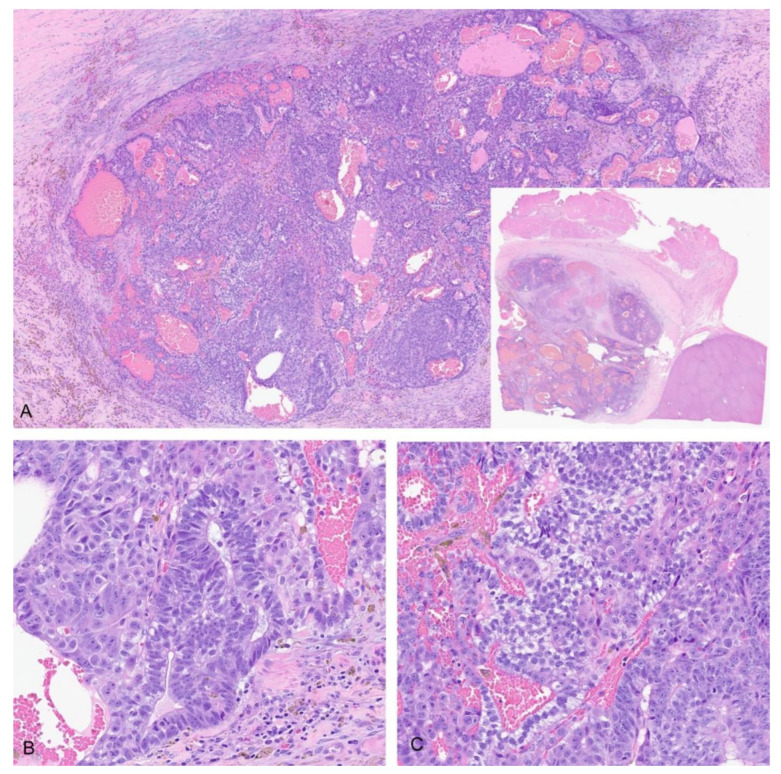
Extrahepatic tumor with teratoid elements, THB23. (**A**) H&E 4×; whole slide image (inset); (**B**) primitive neuroepithelium 20×; (**C**) primitive glands with subnuclear vacuoles 20×.

**Figure 5 cancers-14-06135-f005:**
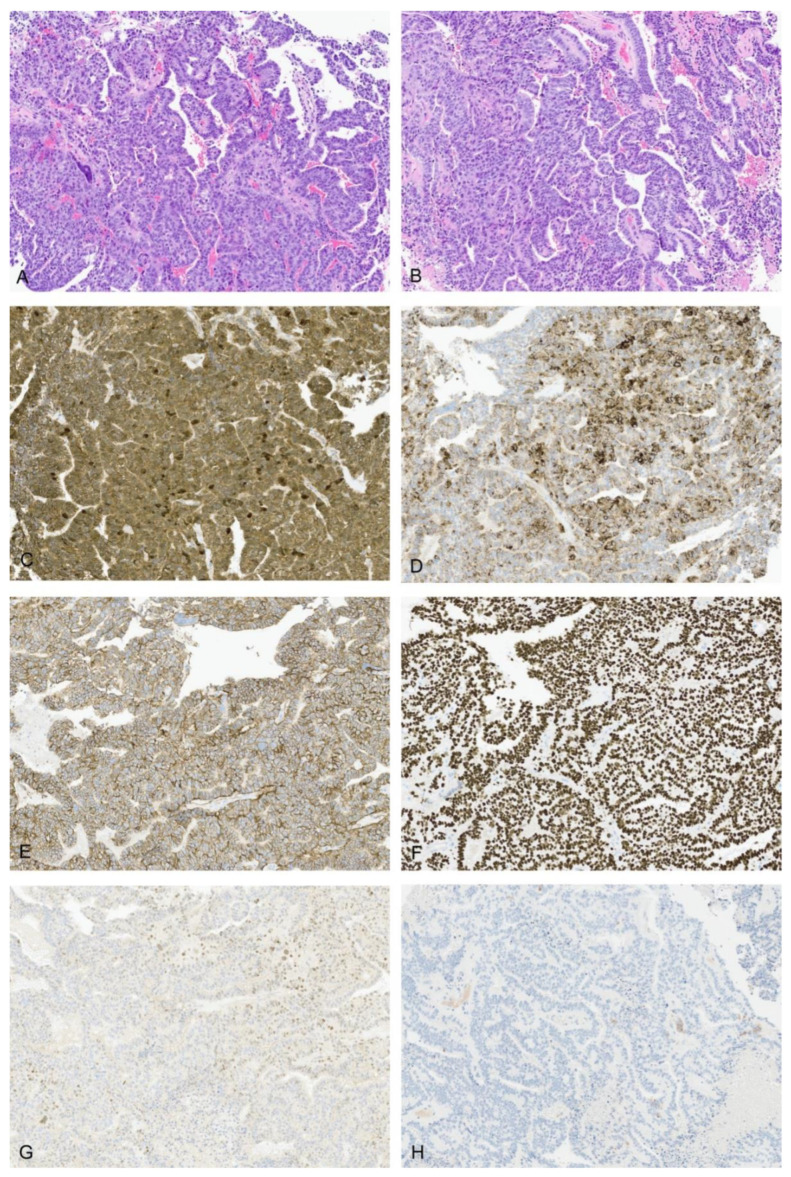
Teratoid HB peritoneal recurrence with tubulopapillary morphology, THB23. H&E 10× (**A**,**B**), beta-catenin IHC 10× (**C**), glypican-3 IHC 10× (**D**), MOC31 IHC 10× (**E**), SALL4 IHC 10× (**F**), PROX-1 IHC 10× (**G**), synaptophysin IHC 10× (**H**).

**Table 1 cancers-14-06135-t001:** Immunohistochemical Stain Information.

IHC Stain	Company	Dilution	Retrieval Process
beta-catenin	Roche ^1^	RTU	Mild CC1
SALL4	Roche ^1^	RTU	Standard CC1
PROX-1	Abcam ^2^	1:400	Standard CC1
synaptophysin	Roche ^1^	RTU	Standard CC1
glutamine synthetase	Roche ^1^	RTU	Mild CC1
MOC31	Roche ^1^	RTU	Mild CC1
glypican-3	Roche ^1^	RTU	Standard CC1
GFAP	Roche ^1^	RTU	Mild CC1
arginase	Roche ^1^	RTU	Standard CC1

RTU—ready to use; CC1—Ventana’s proprietary antigen retrieval solution; ^1^ 1910 Innovation Park Dr. Tucson, AZ, 85755, US; ^2^ Discovery Drive, Cambridge CB2, 0AX, UK.

**Table 2 cancers-14-06135-t002:** Clinicopathologic Features of Teratoid Hepatoblastoma.

Case	Age (mo)	Sex	AFP (ng/mL)	Clinical Features	Surgery	Mets	FU	Pathology
THB1	18	F	559,400		Transplant	N	AWOD	CF, E, mesen, YST, neuroepi, sq
THB2	12	F	N/A		R lobectomy	N	AWOD	F, CF, E, blastema, mesen, neuroepi
THB3	48	F	N/A		Transplant	N	AWOD at 14 years, died 18 years with renal failure	WDF, CF, E, mesen, melanin, sq, PNET-like
THB4	10	M	N/A		Partial hepatectomy	N	AWOD at 25 years	Bx: WDF, E, mesen; Resection: osteoid, mesen, neural, sq, E, F
THB5	12	F	N/A	BWS	L lobectomy	N	AWOD at 9 years	Teratoid glands, CF, E, blastema, bone, neuroepi
THB6	24	F	1,973,536		R lobectomy	N	AWOD at 6 years	Teratoid glands, sq, schwan, pleo F, CF, cholangio
THB7	24	M	1,260,000		Transplant	Y, lung	AWOD at 5 years	Primary: neuroepi, glands, CF, mesen; Mets: fetal pleo (HCC-like)
THB8	36	M	3183	BWS, WHS-endodermal	R lobectomy	N	AWOD at 6 years	Bx: E, CF; Resection: primitive glandular, CF, E
THB9	24	F	140,000		Transplant	N	AWOD at 3 years	Bx: F, E with mucinous matrix; Explant: CF, pleo F, E, primitive glandular, bone
THB10	24	F	262,970		R and L lobe resections	N	AWOD	F, CF, E, mesen, melanin
THB11	12	F	468,590	Prematurity (25 weeks)	Transplant	N	AWOD	Bx: CF, E, WDF, M; Explant: cholangio, teratoid glands, CF, E
THB12	36	M	282,000	Prematurity (27 weeks)	Resection	Y	DOD	Primary: E, mesen, neuroendocrine-like, CF; Mets: YST
THB13	18	F	Elevated	Trisomy 13	Resection	UK	UK	F, CF, E, cholangio, neuroepi, teratoid glands
THB14	18	M	950,000		Extended L hepatectomy, bilateral lung wedge resections	Y, lung	AWD at 1 months fu	Primary: F, CF, E, cholangio, mesen, glial, neuroepi, melanin, ganglion cells, glandular (YST-like), blastema, sq, ductular, rhabdo; Mets: osteoid
THB15	84	M	307,012		Multiple thoracotomies and left hepatic resection	Y, lung	DOD	F, pleomorphic F, E, blastema, primitive glands, macrotrabecular
THB16	24	M	>900,000		Right hemihepatectomy, left liver wedge resections, multiple lung wedge resections	Y, lung	AWD at 2 years fu, recurrence in left liver	Bx: F, CF, E, keratin and sq, pleomorphic F, macrotrabecular, blastema; Resection: CF, E, pleo F, bone, melanin, sq, blastema; Mets: pleo F, HCC-like
THB17	36	M	Elevated		R extended hemihepatectomy	UK	UK	CF, E, blastema, cholangio, mesen, bone, melanin, glial, primtive neuroepi, minor YST
THB18	6	F		BWS (loss of DNA methylation at *DMR2*, VSD, leg-length discrepancy)	R partial hepactectomy	N	AWOD at 5 month fu	Bx: CF, E, blastema, osteoid, neuroepi; Resection: CF, blastema, bone, adipose, skeletal muscle, chondromyxoid, colonic, sq, neural
THB19	24	F	620,752			UK	UK	CF, pleo F, E, macrotrabecular, blastema, neuroepi, melanin
THB20	5	F	UK		Right hepatic lobectomy, lung wedge resections	Y, lung	DOD at 17 months	Primary: F, E, sq, osteoid, pigment, glandular/neuroectodermal, blastema, pleo; Mets: F, E, melanin, sq, ganglion cells, mesen
THB21	11	M	570,000		Extended R hemihepatectomy with caudate lobe resection	N	AWOD at 1 month fu	Bx: F, E, mesen, macrotrabecular, pleo (HCC-like); resection: F, E, mesen, osteoid, bone, sq, melanin, cholangio
THB22	18	F	Elevated		Right hepatic lobectomy	N	AWOD at 17 years	F, E, osteoid, sq, melanin
THB23	48	M	UK		L partial hepatectomy, R liver resection, bilateral lung wedge resections	Y, lung, abdominal nodes, peritoneal, omental	AWD, relapse x2, progression abdominal/pelvic carcinomatosis 3 years after dx	Bx: E; Resection: CF, E, blastema, mesen, osteoid, YST-like, neuroepi, pleo; Lymph node mets: pleo F, E, CF; Peritoneal mets: tubulopapillary YST-like, pleo
THB24	24	M	UK		R hepatic lobectomy	UK	UK	F, CF, cholangio, mesen, smooth muscle, skeletal muscle, mucinous glands, ganglion cells
THB25	24	M	303,000	Precocious puberty, elevated testosterone, 17-OH progesterone, and beta- HCG	L liver lobe resection, adrenal mass excision, splenectomy, partial gastrectomy, omentectomy	Y, omental	UK	CF, E, microcystic YST-like, blastema, glandular, sq, neuroendocrine-like
THB26	72	M	20,603.6		Biopsy	UK	UK	F, E, blastema, mesen, osteoid, neuroglial
THB27	24	M	480,000		Transplant	N	AWOD at 3 years	Primary: F, bone, melanin, sq, glandular/YST-like, cholangio, E; Mets: F, bone, melanin, sq
THB28					Resection			Primitive glands, neuroepi, CF, E, blastema

THB- teratoid hepatoblatoma; Mo- months; FU—follow-up; Mets—metastasis; UK—unknown; AWOD—alive without disease; AWD—alive with disease; DOD—died of disease; BWS—Beckwith–Wiedemann syndrome; WHS—Wolf–Hirschhorn syndrome; Y—yes; N—no; Bx—biopsy; WDF—well-differentiated fetal; F—fetal; CF—crowded fetal; E—embryonal; mesen—mesenchymal; sq—squamous; YST—yolk sac tumor; neuroepi—neuroepithelium; rhabdo—rhabdomyoblastic; pleo—pleomorphic; cholangio—cholangioblastic; HCC—hepatocellular carcinoma; PNET—primitive neuroectodermal tumor.

**Table 3 cancers-14-06135-t003:** Previously reported cases of teratoid HB, mixed HB and teratoma, and mixed HB and YST.

Case	Age	Sex	AFP	Treatment	Clinical Features	Follow Up	Pathology
Case 1 [4]	2.5 yr	M	UK	UK		UK	Mixed HB with striated muscle, respiratory tract tissue, CNS tissue
Case 2 [5]	15 mo	F		Neoadjuvant chemo, surgical resection		UK	Mesenchymal, E, mature intestinal epithelium
Case 3 [6]	8 mo	F	122,100 IU/mL	Partial hepatectomy, second resection, chemo, radiation	Lung mets at autopsy	DOD	Melanin, glandular, ganglion cells, osteoid, E, F (no heterologous elements in second resection or autopsy)
Case 4 [7]	15 mo	M	UK	Chemo, radiation, resection		UK	UK
Case 5 [8]	6 mo	M	UK	None		DOD	F, E, osteoid, glandular YST-like, Schiller-Duval bodies
Cases 6–14 [9]	UK	UK	UK		At least one with mets	6 cases with complete resection (50% survival), 3 cases with incomplete resection (33% survival) overall 44%	Cartilage, skeletal muscle, intestinal-type, squamous, and melanin
Case 15 [10]	15 mo	F	105,000 ng/mL	Partial resection of R lobe of the liver, cytostatic agents		17 mo post-op AWOD	E, F, osteoid, melanin, positive for chromogranin A and serotonin
Case 16 [11]	17 mo	M	77,200 μg/L	Extended L hemihepatectomy, adjuvant chemo		AWOD 37 mo post-op	F, E, sq cysts with pilosebaceous units, eccrine glands, adipose, bone
Case 17 [12]	12 mo	F	UK	UK		UK	osteoid
Case 18 [13]	22 mo	F	418.4 IU/mL; 50,000 IU/mL; 16,686 IU/mL	Neoadjuvant chemo, extended L lobectomy		Died due to septic shock and NEC	E, F, osteoid, neural, melanocytic, endocrine, mucinous
Case 19 [14]	34 yr	M	56,500 μg/L	R lobectomy	Cirrhosis and hepatitis B	DOD, 3 m	E, F, immature glands, primitive mesenchyme, small cell, YST-like, skeletal muscle, neural
Case 20 [15]	8 mo	M	523 IU/mL	L hepatectomy, chemo	No mets	Alive, 12 m	F, E, osteoid, sq, mucinous
Case 21 [16]	3 yr	M	Increased	Neoadjuvant and adjuvant chemo, extended R hemihepatectomy		AWOD 36 mo post-dx	F, E, neuroblastoma-like, rhabdomyoblastic, cystic epithelial structures, sq, osteoid
Case 22 [17]	6 yr 9 mo	F	Normal	R lobectomy and adjuvant chemo	Cushing’s syndrome, ACTH 184–819 pg/mL, hypercalciuria, nephroclacinosis, hypercalcemia, elevated urinary steroids, tumor secreting ACTH, CRH, and PTHrP	AWOD 20 years post-surgery	Infantile sarcoma, typical HB, hormone granules by EM
Case 23 [18]	18 mo	F	UK	Neoadjuvant chemo, liver transplant	Stage IV, lungs mets at presentation	AWOD 1 year post-op	(Pre-tx) keratin/squamous; (explant) squamous, F, E, neuroepithelial, small cells reminiscent of neuroendocrine, spindle cell mesenchyme, melanin
Case 24 [19]	UK	UK	UK	Resection	Lung mets, mucinous differentiation in original HB resection	UK	CDX2 positive mucinous differentiation
Case 25 [20]	15 mo	F	55,300 ng/mL	R lobectomy and adjuvant chemo	Trisomy 13, born at 36 weeks	AWOD at 8 mo after dx	F, E, primitive glandular with well-differentiated adenocarcinoma-like, keratin, osteoid, primitive neuroepithelium, mucinous
Case 26 [21]	18 mo	M	>20,000 ng/mL	Neoadjuvant and adjuvant chemo, R extended liver resection with excision of R hemi-diaphragm	Bx called germ cell tumor, positive surgical resection margin on intitial wedge resection, intravascular cartilage on extended resection	Died due to sepsis 7 days post-op (9 mo after initiation of therapy)	(Extended resection) F, E, osseous metaplasia, cartilage
Case 27 [3]	3 yr	M	282,000 ng/mL	4 rounds of chemotherapy, L extended hepatectomy, 2 additional rounds of chemotherapy	Recurrence post salvage chemo, perirenal mets, retroperitoneal lymphadenopathy	UK	F, E, osteoid, YST-like, neuroendocrine-like
Case 28 [22]	8 mo	F	998 U/mL	Neoadjuvant chemo, extended R hepatectomy	Growing teratoma syndrome (paradoxical increase in tumor despite chemo with normalization of tumor markers)	AWOD 1 mo post-op	(Bx) F, E; (re-bx) teratoid (calling based on sq), osteoid, sq; (resection) ectodermal, mesodermal, endodermal (called teratoma), HB < 2%, adipose, glands CK7 positive
Case 29 [23]	24 mo	M	>830 IU/mL	R hepatectomy, chemo		UK	F, E, melanin
Case 30 [24]	12 mo	F	>400,000 IU/mL	Neoadjuvant chemo, R hepatectomy		Died, air embolism in surgery	F, E, cholangioblastic/primitive glandular, neural, ganglion cells, sq, adipocytes, muscle, cartilage, mucinous, smooth muscle, rhabdomyosarcomatous, osteoid, cartilage

Yr- years; mo- months; UK—unknown; F—fetal; E—embryonal; sq—squamous; YST—yolk sac tumor; DOD—died of disease; AWOD—alive without disease.

## Data Availability

The data presented in this study are available on request from the corresponding author.

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
