# Peer review of "Teratoid Hepatoblastoma—Our Experience"

_cancers, 2022, doi:10.3390/cancers14246135_

Round 1
Reviewer 1 Report
The authors of this study present the largest case series of teratoid hepatoblastomas (HBs) to date, focusing on the spectrum of histologic findings that may be seen, with clinical findings and follow-up reported where available. As the authors note, teratoid HBs represent the rarest subtype of HBs, which are themselves rare, and the literature on teratoid HBs consists predominantly of case reports. This study builds on these case reports, both confirming and further illustrating the histologic spectrum that may be seen in teratoid HBs. Novel findings include reporting of a comprehensive panel of immunohistochemical stains and their results in the various tumor components.
The manuscript is very well-written and clear. Specific recommended changes include:
* Figure 3: Figures B and C appear to be flipped with respect to the legend.
* Table 2: Similar to table 3, it would be helpful to include a column that summarizes/lists the pathologic findings in each case.
Author Response
Point 1: Figure 3: Figures B and C appear to be flipped with respect to the legend.
Response 1: We thank the reviewer for noting this error. The legend for Figure 3 has been corrected to read as follows: “Neuroendocrine-like elements in THB12 and THB25. (A) H&E THB12; (B) Synaptophysin IHC THB12; (C) H&E THB25; (D) Synaptophysin IHC in THB25.”
Point 2: * Table 2: Similar to table 3, it would be helpful to include a column that summarizes/lists the pathologic findings in each case.
Response 2: We thank the reviewer for this recommendation. An additional column summarizing the pathologic findings has been added to Table 2.
Reviewer 2 Report
In the article "Teratoid hepatoblastoma-our experience", the authors compile the histological characteristics of 28 cases of teratoid hepatoblastoma that have been diagnosed in the last 42 years in their hospital. In my opinion, obtaining so many cases of such a rare tumor has a lot of merit and can give a lot of information to other clinicians who may be faced with possible cases of teratoid hepatoblastoma.
The article is well written, easy to follow and the figures are clear and representative. To put a downside, I would have liked to find something more about the treatment received by the patients, especially those diagnosed in the last few years. In this way the article would be of great help not only for the diagnosis but also in the therapy of teratoid hepatoblastoma.
In addition, it would be advisable to clearly indicate what the abbreviations used in the text and tables mean (e.g. RTU, CC1, FU). The legends of the tables are confusing (lines 74 and 75 look like normal text and do not refer to Table 2, and lines 184 and 185 to Table 3). It is also not very intuitive to relate figure legend and its corresponding figure by how it is placed in the document. It would be more convenient if each legend was just below the figure to which it refers.
Apart from those small details, I think it's a good job.
Author Response
Point 1: The article is well written, easy to follow and the figures are clear and representative. To put a downside, I would have liked to find something more about the treatment received by the patients, especially those diagnosed in the last few years. In this way the article would be of great help not only for the diagnosis but also in the therapy of teratoid hepatoblastoma.
Response 1: We thank the reviewer for this comment. We have added the following sentence to the Clinical section of the Results to address this recommendation: “Since the early 2000s, most patients have been treated on standard Children’s Oncology Group protocols for these tumors determined initially by their stage at presentation and ability to resect upfront and more recently on the latest protocol based on PRETEXT staging. Details of all treatments for individual patients are not available for purposes of this study.”
Point 2: In addition, it would be advisable to clearly indicate what the abbreviations used in the text and tables mean (e.g. RTU, CC1, FU).
Response 2: The footers for Table 1 and Table 2 have been updated to include a key for the abbreviations RTU, CC1, and FU.
Point 3: The legends of the tables are confusing (lines 74 and 75 look like normal text and do not refer to Table 2, and lines 184 and 185 to Table 3).
Response 3: We thank the reviewer for this comment. Aside from the footers for Table 2 and Table 3, table legends are not included in the text. A space between the tables and the main text in lines 74-75 and lines 184-185 has been added to improve the readability of the main text and table content.
Point 4: It is also not very intuitive to relate figure legend and its corresponding figure by how it is placed in the document. It would be more convenient if each legend was just below the figure to which it refers.
Response 4: In consideration of this recommendation, all of the figure legends have been moved in the text so that each legend immediately follows the figure to which it refers.